# Association of analgosedation with psychiatric symptoms and health-related quality of life in ARDS survivors: Post hoc analyses of the DACAPO study

**Sebastian Blecha**[1]*, **Florian Zeman**[2], **Magdalena Rohr**[3], **Frank Dodoo-Schittko**[3¤a],
**Susanne Brandstetter**[3¤b], **Christian Karagiannidis**[4], **Christian Apfelbacher**[3¤a],
**Thomas Bein**[5], **for the DACAPO study group**[¶]

**1** Department of Anaesthesiology, University Medical Centre Regensburg, Regensburg, Germany, **2** Centre of Clinical Studies, University Medical Centre Regensburg, Regensburg, Germany, **3** Medical Sociology, Institute of Epidemiology and Preventive Medicine, University of Regensburg, Regensburg, Germany, **4** Department of Pneumology and Critical Care Medicine, Cologne-Merheim Hospital, ARDS and ECMO Centre, Kliniken der Stadt Köln gGmbH, Witten/Herdecke University Hospital, Cologne, Germany, **5** Faculty of Medicine, University of Regensburg, Regensburg, Germany

¤a Current address: Institute of Social Medicine and Health Systems Research, Otto-von-Guericke-University Magdeburg, Magdeburg, Germany
¤b Current address: University Children's Hospital Regensburg (KUNO), University of Regensburg, Regensburg, Germany
¶ Membership of the DACAPO study group is provided in the Acknowledgments.
* Sebastian.Blecha@ukr.de

**Data Availability Statement:** A de-identified data sharing is not possible based on ethical restrictions of the Ethics Committee of the University of

## Abstract

### Background

The acute respiratory distress syndrome (ARDS) is a life-threatening condition with the risk of developing hypoxia and thus requires for invasive mechanical ventilation a long-term analgosedation. Yet, prolonged analgosedation may be a reason for declining health-related quality of life (HRQoL) and the development of psychiatric disorders.

### Methods

We used data from the prospective observational nation-wide ARDS study across Germany (DACAPO) to investigate the influence of sedation and analgesia on HRQoL and the risk of psychiatric symptoms in ARDS survivors 3, 6 and 12 months after their discharge from the intensive care unit (ICU). HRQoL was measured with the Physical and Mental Component Scale of the Short-Form 12 Questionnaire (PCS-12, MCS-12). The prevalence of psychiatric symptoms (depression and post-traumatic stress disorder [PTSD]) was assessed using the Patient Health Questionnaire-9 and the Post-Traumatic Stress Syndrome-14. The associations of analgosedation with HRQoL and psychiatric symptoms were investigated by means of multivariable linear regression models.

Regensburg. The written consent verified by the local Institutional Review Board obtained from the study participants did not include a paragraph for public availability of anonymized patient data. Data are available from the two principal investigators (PIs) Christian Apfelbacher and Thomas Bein (E-mail: Christian.Apfelbacher@med.ovgu.de and/or Thomas.Bein@ukr.de) on request. Our non-author contact with the full data access is the Centre of Clinical Studies of the University Medical Centre Regensburg with the following mail address: zks@ukr.de.

**Funding:** The study is funded by the German Ministry of Education and Research (Bundesministerium für Bildung und Forschung, funding number 01GY1340). The funders had no role in study design, data collection and analysis, decision to publish, or preparation of the manuscript.

**Competing interests:** CK reports personal fees from Maquet, personal fees from Xenios, personal fees from Bayer, non-financial support from Speaker of the German register of ICUs, grants from the German Ministry of Research and Education, during the conduct of the study. All other authors (SB, FZ, FDS, SuB, MB, CA, and TB) declare no conflict of interest.

**Abbreviations:** ARDS, Acute respiratory distress syndrome; BMI, Body mass index; HRQoL, Health-related quality of life; ICU, Intensive care unit; MCS-12, Mental Component Summary; PCS-12, Physical Component Summary; PHQ-9, Patient Health Questionnaire-9; PTSD, Posttraumatic stress disorders; PTSS-14, Post-Traumatic Stress Syndrome 14-Questions Inventory; SAPS II, Simplified Acute Physiology Score; SF -12, Short Form-12 self-report questionnaire; SOFA, Sequential organ failure assessment.

## Results

The data of 134 ARDS survivors (median age [IQR]: 55 [44–64], 67% men) did not show any significant association between analgosedation and physical or mental HRQoL up to 1 year after ICU discharge. Multivariable linear regression analysis (B [95%-CI]) yielded a significant association between symptoms of psychiatric disorders and increased cumulative doses of ketamine up to 6 months after ICU discharge (after 3 months: depression: 0.15 [0.05, 0.25]; after 6 months: depression: 0.13 [0.03, 0.24] and PTSD: 0.42 [0.04, 0.80)]).

## Conclusions

Up to 1 year after ICU discharge, analgosedation did not influence HRQoL of ARDS survivors. Prolonged administration of ketamine during ICU treatment, however, was positively associated with the risk of psychiatric symptoms. The administration of ketamine to ICU patients with ARDS should be with caution.

## Trial registration

Clinicaltrials.gov: NCT02637011 (Registered 15 December 2015, retrospectively registered).

## Background

The acute respiratory distress syndrome (ARDS) is a life-threatening condition, which is characterised by either direct or indirect damage to the lung parenchyma, often resulting in critical hypoxemia or hypercapnia, or both [1]. In the initial phase of ARDS, patients require long-term analgosedation for invasive mechanical ventilation. In the current context, patients with severe COVID-19-associated ARDS are usually ventilated for a mean duration of 13 to 16 days [2, 3]. Prolonged sedation during intensive care therapy may be a potential risk factor for the development of psychiatric disorders and delirium [4, 5].

Additionally, ARDS survivors have a substantial risk of developing anxiety, depression and symptoms of posttraumatic stress disorder (PTSD) as well as of decreased health-related quality of life (HRQoL) [6, 7]. A recent study of 114 COVID-19-associated ARDS survivors in Italy showed symptoms of depression in 9% of patients, of anxiety in 10% and of PTSD in 4% [8]. A study of 113 COVID-19-associated ARDS survivors in Spain showed that over 90% of patients had developed one or more symptoms of a mental disorder [9].

In the current study, we assessed the type and dosage of sedation and analgesia administered during intensive care treatment and the influence of analgosedation on HRQoL and psychiatric disorders up to 1 year after ARDS survival. We hypothesised that analgesics and sedation drugs and their administered doses during ICU treatment increased the risk of psychiatric symptoms and reduced HRQoL of long-term survivors of ARDS.

## Methods

### Study design

The extent and duration of analgosedation and co-medications of ARDS survivors was recorded in the context of a large prospective nation-wide cohort study across Germany (DACAPO study, ClinicalTrials.gov Identifier: NCT02637011) [10]. The DACAPO study was focused on the influence of the quality of care and individual patient characteristics on

HRQoL and the time point of returning to work of ARDS survivors. Ethical approval was obtained from the ethics committee of the University of Regensburg (file number: 13–101–0262) and additionally from the ethics committees overseeing the respective study sites [11]. The baseline characteristics and profile of the cohort have been described in more detail elsewhere [12]. ARDS survivors were asked to complete comprehensive self-report questionnaires 3, 6 and 12 months after their discharge from the intensive care unit (ICU).

## Sample

**Fig 1** gives an overview of the sample size at different time points of the study. Written informed consent was obtained from 1,225 patients with ARDS, who had been treated at one of 61 ICUs across Germany between September 2014 and April 2016. The study included adults with the diagnosis of ARDS (according to the Berlin definition [13]). Patients or their caregivers or legal guardians were approached during the patients' ICU stay and asked to provide written informed consent. In cases in which caregivers or legal guardians consented to a patient's participation in the study, the patient had to confirm this preliminary consent at discharge from the ICU.

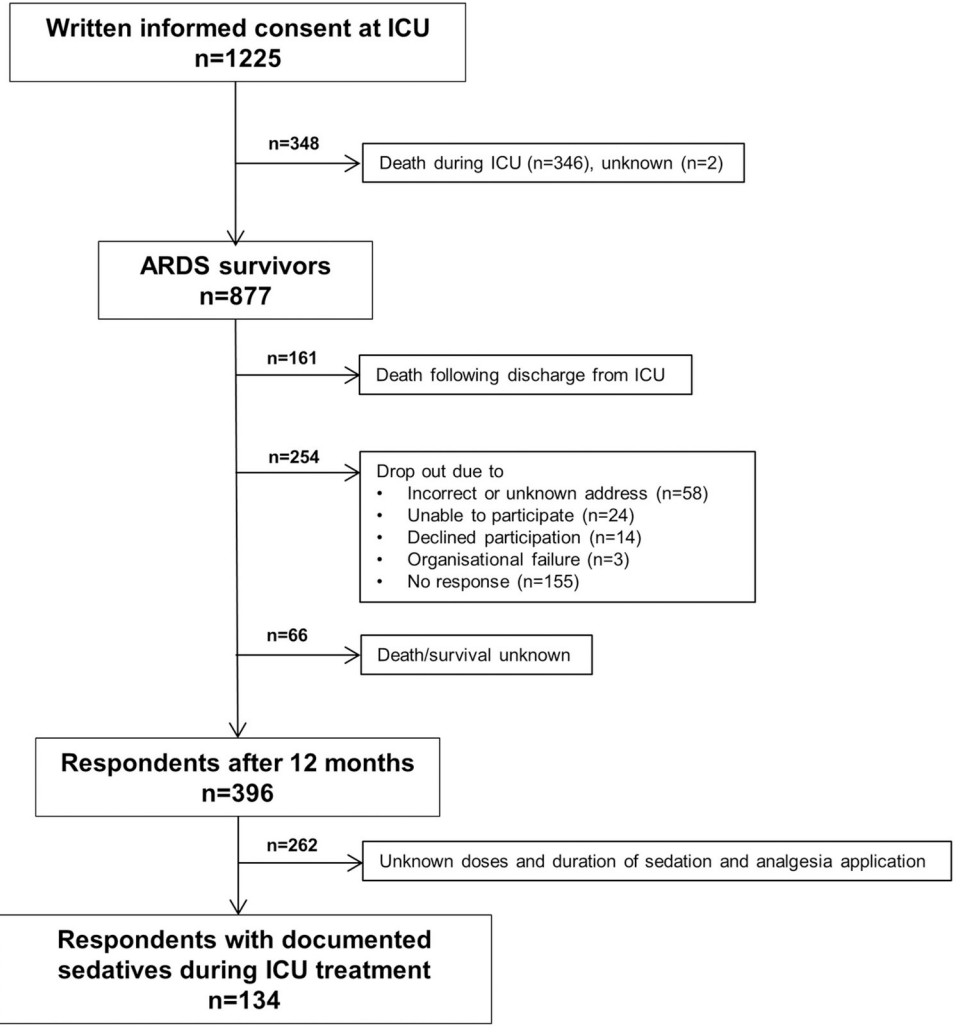

**Fig 1. Flow chart of patients with ARDS throughout the study.**

Out of 877 ICU survivors, 396 (45.2%) had returned the questionnaire at 12 months. The most frequent reason for dropping out of the study was death after discharge from the ICU (N = 161). Other reasons included the inability to complete the questionnaire (insufficient knowledge of German or incapability due to morbidity), absence of a person who could provide proxy reports, withdrawal of consent or an invalid address. Furthermore, the doses and duration of analgosedation could not be assessed in 262 ARDS survivors.

### Data collection and measuring instruments

Characteristics of patient disease and treatment as well as information on ICU discharge were reported by study physicians or physicians from the participating ICUs using the electronic data capture system OpenClinica (OpenClinica, LLC; https://www.openclinica.com/). The type of analgesia and sedation, duration and cumulative doses of drugs were extracted and transmitted from the patient data management system of nine recruiting study centres (8 University hospitals and one hospital for basic and standard care). Information on HRQoL and psychiatric disorders was assessed by means of self-report questionnaires at 3, 6 and 12 months after discharge from the ICU. ARDS survivors completed the Short Form-12 self-report questionnaire (SF-12) as a measure of HRQoL and questionnaires on depressive symptoms and post-traumatic stress disorder (PTSD). The Patient Health Questionnaire-9 [PHQ-9] was used to detect symptoms of depression and the Post-Traumatic Stress Syndrome 14-Questions Inventory [PTSS-14] to detect PTSD [14]. For the screening tools, the cut-off values for being at risk of depression were defined as PHQ-9 $\geq 5$ and for PTSD as PTSS-14 $\geq 45$ [15, 16]. The published scoring algorithm of SF-12 resulted in the Physical Component Summary (PCS-12) and the Mental Component Summary (MCS-12) scores. Scores range from 0 to 100 (higher values indicate better HRQoL); a score of 50 represents the mean value for the general population (German norm values [17, 18]).

### Statistical analyses

All patients of the prospective DACAPO study with data on analgosedation were used for this supplementary analysis. Thus, no more than n = 134 patients were available. Nevertheless, this sample is sufficient for the first exploratory analyses regarding our additional questions. As a rule of thumb, a minimum of 10–12 cases is needed per variable for a linear regression model. In our multiple linear regression model with eight independent variables, we needed at least 80 cases for the first exploratory analyses. Data are shown as mean ± SD or median (IQR) for continuous variables and as absolute and relative frequencies for categorical variables. To assess which analgosedation has a potential impact on HRQoL and the risk of psychiatric symptoms after 3, 6 and 12 months, bivariate models (including type of analgosedation yes/no plus cumulative doses) were calculated for each type of analgosedation and each endpoint. For each endpoint, all significant results were included in a multiple linear regression model, which was further adjusted for the known risk factors age, sex, Simplified Acute Physiology Score II (SAPS II) and length of the ICU stay. The regression coefficient B for the linear regression models was calculated as effect estimate accompanied by the corresponding 95%-confidence interval (95%-CI). A p-value $<0.05$ was considered significant. Due to the exploratory nature of this study, no adjustments for multiple comparisons were used. All analyses were performed using the software R (Version 3.5.1, www.r-project.org).

### Results

Out of 396 responding ARDS survivors, 134 patients were analysed 12 months after their ICU discharge (**Fig 1**). The patient characteristics are shown in Table 1. Median age at ICU

**Table 1. Patient characteristics of ARDS survivors with documented analgosedation (n = 134).**

| Patient characteristics | | Missing (n) |
|---|---|---|
| **Sex** | | |
| Men, n (%) | 90 (67) | 0 |
| Women, n (%) | 44 (33) | |
| **Age,** years, median (IQR) | 55 (44–64) | 0 |
| **BMI (kg/m$^2$),** median (IQR) | 26 (24–30) | 0 |
| **Severity of ARDS[a]** | | |
| Mild, n (%) | 24 (17.9) | 0 |
| Moderate, n (%) | 75 (56.0) | |
| Severe, n (%) | 35 (26.1) | |
| **SAPS II[b],** median (IQR) | 38 (30–46) | 5 |
| **SOFA score[a],** median (IQR) | 8 (6–11) | 9 |
| **Duration of ICU stay,** days, median (IQR) | 23.5 (15–35) | 0 |
| **Tracheotomy during ICU stay** | | |
| Yes, n (%) | 61 (46) | 1 |
| No, n (%) | 72 (54) | |

Notes: ARDS: acute respiratory distress syndrome, BMI: body mass index, ICU: intensive care unit, SAPS II: Simplified Acute Physiology Score II, SOFA: Sequential Organ Failure Assessment

[a]measured at the time of ARDS diagnosis

[b]measured 24 hours after ICU admission

admission was 55 years, two-thirds of the 12-month respondents were men and median length of the ICU stay was 23.5 days. The majority of patients had moderate or severe ARDS (82.1%). Tracheostomy was performed in 46% of ARDS survivors after a median time of 11 days (IQR: 7–16).

The drugs used for analgesia and sedation during ICU treatment in ARDS survivors are shown in Table 2. In the following, the number of sedative and analgesic agents that patients received during ICU treatment is listed (based on Table 2): 12 patients (8.9%) got a sixfold, 28 patients (20.9%) fivefold, 38 patients (28.4%) a fourfold, 34 patients (25.4%) a threefold, 19 patients (14.2%) twofold and only 3 patients (2.2%) single analgosedation. The most common type of analgosedation was the combination of intravenously administered propofol and sufentanil. Sedation was often deepened with clonidine (73.1%) and/or midazolam (59.0%). Ketamine or esketamine was administered in 32.1% of patients. Thirty-two of 39 patients (82.1%) with ketamine application and all patients with esketamine additionally received

**Table 2. Total cumulative dose and duration of sedation and analgesia in ARDS survivors (N = 134).**

| Sedation and analgesic agents | n (%) | Total cumulative dose | Duration of sedation and analgesia |
|---|---|---|---|
| | | mg, median (IQR) | days, median (IQR) |
| Propofol | 129 (96.3) | 21355 (11786–39177) | 7 (4–10) |
| Sufentanil | 124 (92.5) | 5121 (23–18386) | 12 (7–20) |
| Midazolam | 79 (59.0) | 1084 (295–4100) | 6 (3–14) |
| Ketamine | 39 (29.1) | 8830 (2247–31034) | 3 (1–7) |
| Esketamine | 4 (3.0) | 5225 (164–10260) | 5 (2–8) |
| Clonidine | 98 (73.1) | 7 (3–20) | 5 (2–10) |
| Dexmedetomidine | 20 (14.9) | 0.21 (0.03–2.62) | 4 (2–8) |
| Isoflurane | 8 (6.0) | - | 8 (4–15) |

midazolam. Inhalative sedation with isoflurane administered via the AnaConDa® system was used in only 6% of patients.

HRQoL and symptoms of psychiatric disorders of ARDS survivors 3, 6 and 12 months after ICU discharge are shown in Table 3. In 84 of 134 ARDS survivors, for whom medication was documented during ICU therapy, the values of SF-12, PHQ-9, and PTSS-14 could be matched. According to SF-12 measurement, ARDS survivors had lower HRQoL (<50) across all follow-up time points compared to the general population norm [18]. Additionally, the mean increased PHQ-9 values show symptoms of depression across all follow-up time points. 20.2% of patients reported symptoms of PTSD (cut-off scores for symptoms of PTSD: ≥45) and 54.2% symptoms of depression (PHQ-9 score ≥5) at the 1-year follow-up, respectively.

Analysing the impact of analgosedation of all given drugs (propofol, midazolam, sufentanil, [es-]ketamine, clonidine, dexmedetomidine and isoflurane) on HRQoL and symptoms of psychiatric disorders during the follow-up time points, only the cumulative doses of ketamine and midazolam showed a significant linear relationship to PHQ-9 and PTSS-14 in the bivariable linear regression models. The relationship of ketamine to PHQ-9 and PTSS-14 scores is shown in **Fig 2** and that of midazolam to PHQ-9 and PTSS-14 scores in **Fig 3**. HRQoL (PCS-12/MCS-12) was not affected by analgesia and sedation at any follow-up time point.

Multivariable linear regression analysis only established the cumulative dose of ketamine as a significant predictor for depression and PTSD (Table 4 **and S1 Table**). Higher doses of ketamine increased the risk of psychiatric symptoms in ARDS survivors up to 6 months after ICU discharge.

**Table 3. HRQoL and symptoms of psychiatric disorders in ARDS survivors after 3, 6 and 12 months (N = 84).**

| | Measured value | Patients whose scores differed from the cut-off values | Missing (n) |
|---|---|---|---|
| | mean (±SD) | n (%) | |
| **3 months after ICU discharge** | | | |
| PCS-12[a] | 36.6 (±9.5) # | # 64 (87.7) | 11 |
| MCS-12[a] | 43.7 (±12.1) # | # 46 (63.0) | 11 |
| PHQ-9[a] | 5.6 (±4.7)* | * 38 (45.8) | 1 |
| PTSS-14[a] | 29.1 (±13.5) | * 12 (14.5) | 1 |
| **6 months after ICU discharge** | | | |
| PCS-12[a] | 40.2 (±11.8) # | # 47 (75.8) | 22 |
| MCS-12[a] | 42.1 (±14.4) # | # 40 (64.5) | 22 |
| PHQ-9[a] | 6.1 (±5.1)* | * 45 (54.2) | 1 |
| PTSS-14[a] | 32.2 (±17.7) | * 16 (20.0) | 4 |
| **12 months after ICU discharge** | | | |
| PCS-12[a] | 42.7 (±11.7) # | # 46 (63.9) | 12 |
| MCS-12[a] | 45.6 (±13.9) # | # 38 (52.8) | 12 |
| PHQ-9[a] | 5.8 (±5.2)* | * 45 (54.2) | 1 |
| PTSS-14[a] | 31.5 (±16.8) | * 17 (20.2) | 0 |

Notes: MCS-12: mental component scale of short-form 12 questionnaire; PCS-12: physical component scale of short-form 12 questionnaire; PHQ-9: Patient Health Questionnaire-9, PTSS-14: Post-Traumatic Stress Syndrome 14-Questions Inventory

[a]health-related quality of life and symptoms of psychiatric disorders (depression, post-traumatic stress disorder) were assessed according to the results of patient self-reported questionnaires

#PCS-12<50 and MCS-12<50 represents worse HRQoL compared to that of the general (German) population

*cut-off values for being at risk of depression and PTSD: PHQ-9-score ≥5 and PTSS-14-score ≥45

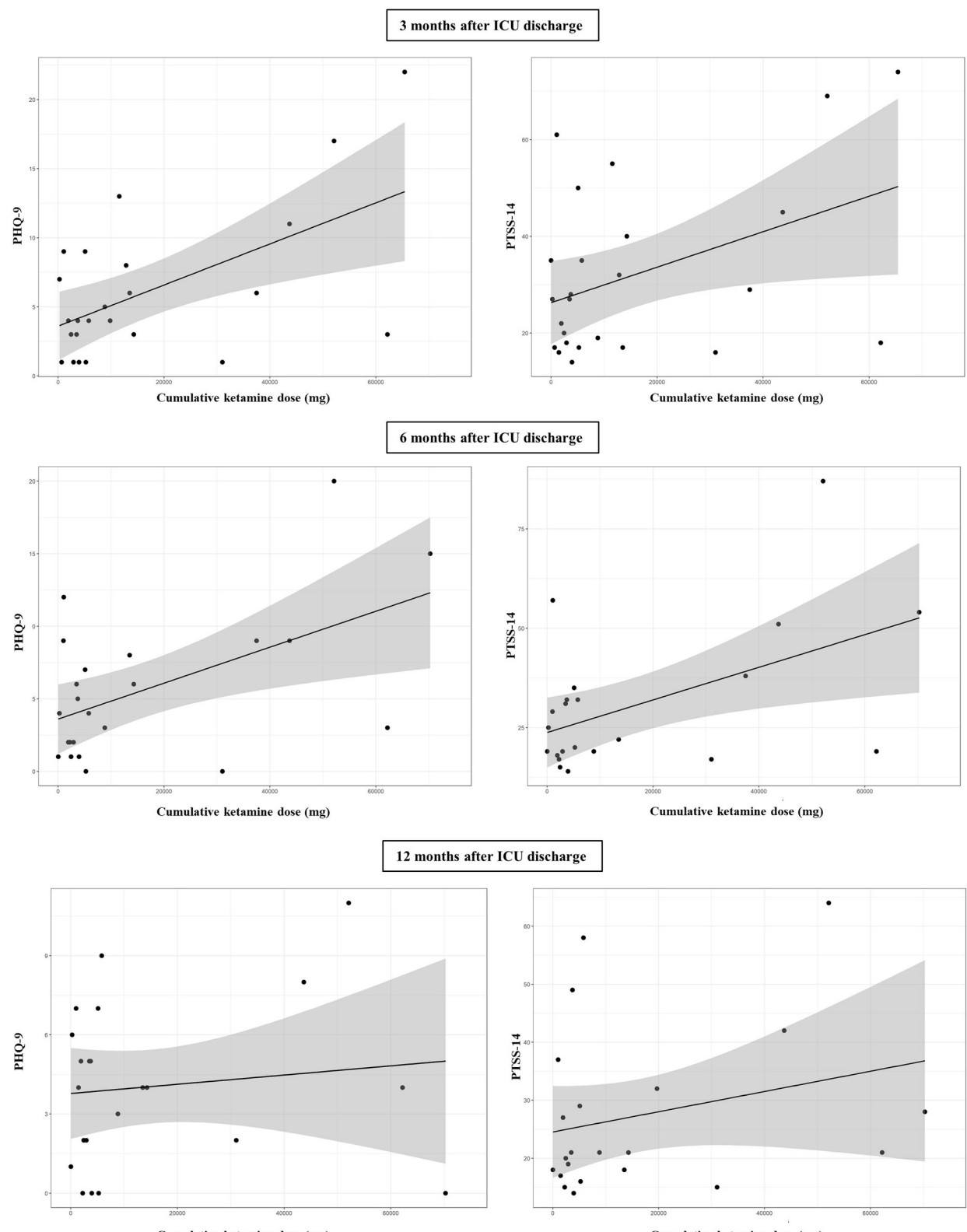

**Fig 2. Relationship between cumulative dose of ketamine and psychiatric symptoms in ARDS survivors 3, 6 and 12 months after ICU discharge.**

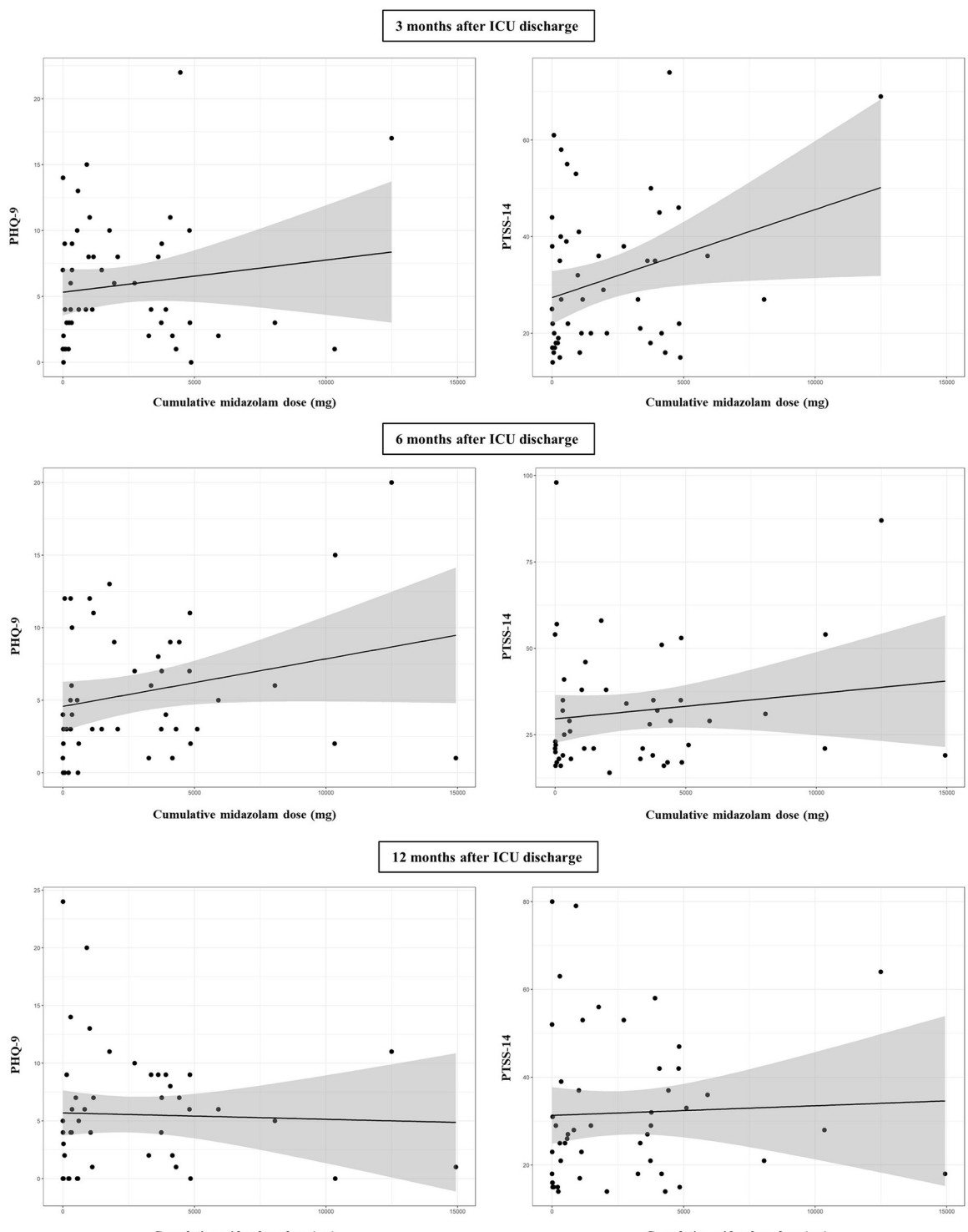

**Fig 3. Relationship between cumulative dose of midazolam and psychiatric symptoms in ARDS survivors 3, 6 and 12 months after ICU discharge.**

**Table 4. Multiple linear regression models of the influence of cumulative ketamine doses on psychiatric symptoms in ARDS survivors 3, 6 and 12 months after ICU discharge.**

| Independent variables[#] | PHQ-9[a] | | PTSS-14[a] | |
|---|---|---|---|---|
| | B (95%-CI) | P-value | B (95%-CI) | P-value |
| **3 months after ICU discharge** | | | | |
| Ketamine | −1.5 (-4.3, 1.3) | 0.297 | −0.12 (−7.74, 7.50) | 0.975 |
| Cumulative ketamine dose** | 0.15 (0.05, 0.25) | **0.004**[*] | 0.24 (−0.05, 0.53) | 0.102 |
| Midazolam | 0.60 (−1.83, 3.02) | 0.625 | 2.53 (−4.27, 9.33) | 0.461 |
| Cumulative midazolam dose** | −0.09 (−0.61, 0.44) | 0.742 | 0.96 (−0.69, 2.61) | 0.248 |
| **6 months after ICU discharge** | | | | |
| Ketamine | −2.8 (−5.8, 0.19) | 0.066 | −9.94 (−20.86, 0.99) | 0.074 |
| Cumulative ketamine dose** | 0.13 (0.03, 0.24) | **0.014**[*] | 0.42 (0.04, 0.80) | **0.029**[*] |
| Midazolam | −2.0 (−4.7, 0.67) | 0.138 | −2.65 (−12.31, 7.02) | 0.587 |
| Cumulative midazolam dose** | 0.29 (−0.20, 0.78) | 0.241 | 0.63 (−1.14, 2.40) | 0.479 |
| **12 months after ICU discharge** | | | | |
| Ketamine | −2.68 (−5.97, 0.62) | 0.110 | −8.58 (−19.53, 2.38) | 0.123 |
| Cumulative ketamine dose** | 0.03 (−0.08, 0.15) | 0.572 | 0.15 (−0.24, 0.53) | 0.453 |
| Midazolam | 0.04 (−2.80, 2.88) | 0.978 | −0.62 (−10.00, 8.75) | 0.895 |
| Cumulative midazolam dose** | 0.09 (−0.47, 0.64) | 0.761 | 0.46 (−1.36, 2.29) | 0.616 |

Notes

[*]p <0.05

**per 1000 mg

[#]further adjusted for age, sex, SAPS, days on ICU; B: regression coefficient, 95%-CI: 95% confidence interval, PHQ-9: Patient Health Questionnaire-9; PTSS-14: Post-Traumatic Stress Syndrome 14-Questions Inventory

[a]symptoms of psychiatric disorders (depression and post-traumatic stress disorder) were diagnosed according to the results of patient self-reported questionnaires

## Discussion

This study analysed the influence of sedation and analgesia on HRQoL and the prevalence of psychiatric symptoms in 134 ARDS survivors up to 1 year after ICU discharge. The main findings were: 1) Increased cumulative ketamine doses for analgosedation during ICU treatment were significantly associated with a higher prevalence of psychiatric symptoms 3 and 6 months after ICU discharge and 2) Analgosedation was not significantly associated with physically (PCS-12) or mentally impaired quality of life (MCS-12).

### Analgosedation during ICU treatment

At the onset of ARDS, deep analgosedation is often needed. After initial stabilisation during the recovery phase, a protocol-based sedation regimen emphasising light or no sedation, daily spontaneous awakening and breathing periods as well as physiotherapy may improve short-term outcome and decrease long-term complications including the risk of cognitive decline and psychiatric symptoms [19, 20].

In our study, the most common type of analgosedation during ICU treatment was the combination of intravenously administered propofol and sufentanil, often supplemented or replaced with midazolam, clonidine or dexmedetomidine. Ketamine was administered in nearly one third of the patients. In a recent study by Ren and colleagues, all 42 patients with ARDS who received venovenous extracorporeal membrane oxygenation were given midazolam combined with sufentanil or remifentanil; in 69%, analgosedation was supplemented with propofol and in 25% with dexmedetomidine [21].

## HRQoL and psychiatric symptoms in ARDS survivors

The literature for analysing the association between analgosedation and the development of psychiatric symptoms in ARDS survivors is very scarce. Stevenson and colleagues found that 38% of 152 ARDS survivors had a positive screening test for general anxiety 3 months after ICU discharge, and higher pre-ICU body mass index and psychiatric comorbidity were negatively associated with the development of general anxiety, but without any association of daily administered analgesics, sedatives and steroids with general anxiety symptoms [22]. Bienvenu et al. showed in a 5-year follow-up study that generally 32% of ARDS survivors had prolonged symptoms of depression, 38% of anxiety and 23% of PTSD [6]. Furthermore, the Physical and Mental Component Score often shows decreased HRQoL in ARDS survivors in the long-term follow up [23]. In our study, ARDS survivors had lower HRQoL measured with the PSC-12 and MCS-12 at each follow-up time point compared to the score validated from the general population [17]. Except for ketamine, analgosedation had no impact on HRQoL or the development of psychiatric symptoms. In this post hoc-analysis, 20.2% of patients had symptoms of PTSD and 54.2% symptoms of depression at the 1-year follow-up.

## Indications and risks of ketamine and midazolam administration

Ketamine is an established sedative with a unique mechanism of action and side effect profile. The primary mechanism of action is antagonism of the N-methyl-D-aspartate receptor, which appears to be responsible for the amnestic and analgesic effects. In addition, ketamine stimulates noradrenergic neurons and inhibits catecholamine uptake, which may preserve cardiovascular stability. Ketamine also enhances the descending inhibiting serotoninergic pathway and exerts anti-depressive effects [24]. This is the reason why ketamine is commonly used in patients with major depressive disorders, bipolar depression and treatment-resistant depression [25–27].

A recent prospective study including more than 1000 critically ill surgical and medical patients requiring analgosedation suggested that ketamine may also increase the risk of incident delirium [28]. However, another study including 172 ICU patients showed adjunctive continuous infusion with ketamine to be safe and effective for sedation without any increased risk of delirium. Additionally, in that study, ketamine was associated with reduced norepinephrine requirement, days of benzodiazepine administration and continuous opioid infusion [29]. In our study, higher cumulative doses of ketamine increased the risk of psychiatric symptoms in ARDS survivors up to 6 months after ICU discharge. Currently, it is unknown why ketamine increased the risk of psychiatric symptoms in the short follow-up outcome and not in the long-term follow-up.

For decades, midazolam as a γ-aminobutyric acid (GABA) receptor agonist has been one of the most commonly administered sedative drugs for ICU patients [30, 31]. However, a benzodiazepine dominant sedation strategy (e.g. with midazolam) is associated with an increased risk of developing delirium, prolonged duration of invasive mechanical ventilation and ICU length of stay [32–34]. ICU delirium is common and may increase mortality risk and contribute to long-term cognitive impairment [35, 36]. In our study, higher cumulative doses of midazolam did not affect the HRQoL and risk of psychiatric symptoms in ARDS survivors.

## Strengths and limitations

The strengths of the present study are its prospective design with three follow-up time points, the large number of included patients with ARDS from hospitals across Germany and the detailed collection of data on HRQoL, psychiatric symptoms and individual patient characteristics. Despite our best efforts to follow up each patient, the number of drop-outs and loss of

analgosedation data acquirement was rather high, which may have resulted in selection bias limiting this study. In this post-hoc data collection only a selection of sedatives and analgetics could be used for reasons of practicability. The instruments used for screening mental disorders do not allow making diagnoses such as major depression disorders and PTSD; thus, only symptoms or the risk of being affected by such a disorder were recorded.

## Conclusion

In summary, prolonged administration of ketamine during ICU treatment was positively associated with the risk of psychiatric symptoms. The analgesia and sedation did not influence HRQoL of ARDS survivors up to 1 year after ICU discharge. The administration of ketamine to ICU patients with ARDS should be with caution or avoided as alternative sedative and analgesic drugs are available. Larger prospective studies are needed to analyse the influence of analgosedation and notably of ketamine during ICU treatment on the development of psychiatric symptoms in ARDS survivors.

## Supporting information

**S1 Table. Multiple linear regression models of the influence of cumulative analgosedation (without ketamine and midazolam) on psychiatric symptoms in ARDS survivors 3, 6 and 12 months after ICU discharge.** [*]p <0.05; [**]per 1000 mg; [#]further adjusted for age, sex, SAPS, days on ICU; B: regression coefficient, 95%-CI: 95% confidence interval, PHQ-9: Patient Health Questionnaire-9; PTSS-14: Post-Traumatic Stress Syndrome 14-Questions Inventory; [a]symptoms of psychiatric disorders (depression and PTSD) were diagnosed according to the results of patient self-reported questionnaires.
(PDF)

## Acknowledgments

We are indebted to the highly committed intensivists and study assistants across Germany who had recruited patients for the DACAPO study:

Aachen, Aachen University Hospital RWTH Aachen, Department of Anaesthesiology (PD Dr. Johannes Bickenbach, Dr. Thorben Beeker, PD Dr. Tobias Schürholz, Jessica Pezechk); Amberg, Klinikum Amberg, Department for Anaesthesiology (Dr. Jens Schloer); Augsburg, Klinikum Augsburg (Dr. Ulrich Jaschinski, Ilse Kummer); Bamberg, Sozialstiftung Bamberg Hospital, Department for Anaesthesiology (Dr. Oliver Kuckein); Berlin, Charité—University Medicine Berlin, Department of Anaesthesiology and Intensive Care Medicine (PD Dr. Steffen Weber-Carstens, Dr. Anton Goldmann, Dr. Stefan Angermair, Krista Stoycheva); Berlin, HELIOS Klinikum Berlin-Buch, Department of Intensive Care Medicine (Prof. Dr. Jörg Brederlau, Nadja Rieckehr, Gabriele Schreiber, Henriette Haennicke); Bielefeld, Ev. Krankenhaus Bielefeld. Department of Anaesthesiology, Intensive Care Medicine, Emergency Medicine and Pain Therapy (Dr. Friedhelm Bach, Dr. Immo Gummelt, Dr. Silke Haas, Catharina Middeke, Dr. Ina Vedder, Marion Klaproth); Bochum, Ruhr University Bochum, Department of Anaesthesiology (Prof. Dr. Michael Adamzik, Dr. Jan Karlik, Dr. Stefan Martini, Luisa Robitzky); Bonn, University Hospital Bonn, Department of Anaesthesiology and Intensive Care Medicine (Prof. Dr. Christian Putensen, Dr. Thomas Muders, Ute Lohmer); Bremen, Klinikum Bremen-Mitte, Department of Anaesthesiology (Prof. Dr. Rolf Dembinski); Deggendorf, Medical Centre, Department of Anaesthesiology and Intensive Care Medicine (Dr. Petra Schäffner, Dr. Petra Wulff-Werner); Dortmund, Klinikum Dortmund, Department of Critical Care Medicine (Elke Landsiedel-Mechenbier, Daniela Nickoleit-Bitzenberger, Ann-Kathrin Silber); Dresden,

University Hospital Dresden Carl Gustav Carus, Department of Anaesthesiology and Intensive Care Medicine (Prof. Dr. Maximilian Ragaller, Prof. Dr. Marcello Gama de Abreu, Alin Ulbricht, Linda Reisbach); Frankfurt am Main, University Hospital Frankfurt, Department of Anaesthesiology, Intensive Care Medicine and Pain Therapy (Prof. Dr. Kai Zacharowski, Prof. Dr. Patrick Meybohm, Simone Lindau, Haitham Mutlak); Freiburg, University Medical Center Freiburg, Department of Anaesthesiology and Critical Care Medicine Freiburg (Prof. Dr. Alexander Hötzel, Dr. Johannes Kalbhenn); Freising, Klinikum Freising, Department of Anaesthesiology (Dr. Christoph Metz, Dr. Stefan Haschka); Göppingen, Klinik am Eichert, ALB FILS Kliniken, Department of Anaesthesiology and Intensive Care (Dr. Stefan Rauch); Göttingen, University Medical Center, Department of Anaesthesiology, Emergency and Intensive Care Medicine (Prof. Dr. Michael Quintel, Dr. Lars-Olav Harnisch, Dr. Sophie Baumann, Andrea Kernchen); Greifswald, University Medicine Greifswald, Department of Internal Medicine B (Dr. Sigrun Friesecke, Sebastian Maletzki); Hamburg, University Hospital Hamburg-Eppendorf, Department of Intensive Care Medicine, Centre of Anaesthesiology and Intensive Care Medicine (Prof. Dr. Stefan Kluge, Dr. Olaf Boenisch, Dr. Daniel Frings, Birgit Füllekrug, Dr. Nils Jahn, Dr. Knut Kampe, Grit Ringeis, Brigitte Singer, Dr. Robin Wüstenberg); Hannover, Hannover Medical School, Department of Anaesthesiology and Intensive Care Medicine (Dr. Jörg Ahrens, Dr. Heiner Ruschulte, Dr. Andre Gerdes, Dr. Matthias Groß); Hannover, Hannover Medical School, Department of Respiratory Medicine (Dr. Olaf Wiesner, Aleksandra Bayat-Graw); Heidelberg, University of Heidelberg, Department of Anaesthesiology (Dr. Thorsten Brenner, Dr. Felix Schmitt, Anna Lipinski); Herford, Klinikum Herford, Clinic for Anaesthesiology, Surgical Intensive Care Medicine, Emergency Care Medicine, Pain Management (Prof. Dr. Dietrich Henzler, Dr. Klaas Eickmeyer, Dr. Juliane Krebs, Iris Rodenberg); Homburg, Homburg University Medical Centre, Department of Anaesthesiology, Intensive Care and Pain Medicine (Dr. Heinrich Groesdonk, Kathrin Meiers, Karen Salm, Prof. Dr. Thomas Volk); Ibbenbüren, Ibbenbüren General Hospital, Division of Thoracic Surgery and Lung Support (Prof. Dr. Stefan Fischer, Basam Redwan); Immenstadt, Kempten-Oberallgaeu Hospitals, Clinic for Pneumology, Thoracic Oncology, Sleep- and Respiratory Critical Care (Dr. Martin Schmölz, Dr. Kathrin Schumann-Stoiber, Simone Eberl); Ingolstadt, Klinikum Ingolstadt, Department of Anaesthesiology and Critical Care Medicine (Prof. Dr. Gunther Lenz, Thomas von Wernitz-Keibel, Monika Zackel); Jena, Jena University Hospital, Department of Anaesthesiology and Intensive Care Therapy (Dr. Frank Bloos, Dr. Petra Bloos, Anke Braune, Anja Haucke, Almut Noack, Steffi Kolanos, Heike Kuhnsch, Karina Knuhr-Kohlberg); Kassel, Klinikum Kassel, Department of Anaesthesiology (PD Dr. Markus Gehling); Kempten, Klinikum Kempten-Oberallgäu gGmbH, Department for Anaesthesia and Operative Intensive Care (Prof. Dr. Mathias Haller, Dr. Anne Sturm, Dr. Jannik Rossenbach); Kiel, University Medical Center Schleswig-Holstein, Campus Kiel, Department of Anaesthesiology and Intensive Care Medicine (Dr. Dirk Schädler, Stefanie D'Aria); Köln, Cologne-Merheim Hospital, Department of Pneumology and Critical Care Medicine (Prof. Dr. Christian Karagiannidis, Dr. Stephan Straßmann, Prof. Dr. Wolfram Windisch); Köln, University Hospital of Cologne, Department of Anaesthesiology and Intensive Care Medicine (Prof. Dr. Thorsten Annecke, PD Dr. Holger Herff); Langen, Asklepios Kliniken Langen-Seligenstadt, Department of Anaesthesiology and Intensive Care Medicine (Dr. Michael Schütz); Leipzig, University of Leipzig, Department of Anaesthesiology and Intensive Care Medicine (PD Dr. Sven Bercker, Hannah Reising, Mandy Dathe, Christian Schlegel); Ludwigsburg, Klinikum Ludwigsburg, Academic Teaching Hospital, University of Heidelberg, Department of Anaesthesiology (Katrin Lichy); Ludwigshafen, Klinikum Ludwigshafen, Department of Anaesthesiology and Intensive Care Medicine (Prof. Dr. Wolfgang Zink, Dr. Jana Kötteritzsch); Mainz, University Medical Centre Mainz, Department of Anaesthesiology (Dr. Marc Bodenstein, Susanne Mauff, Peter Straub);

Magdeburg, Magdeburg University Medical Centre, Department of Anaesthesiology and Intensive Care Medicine (Dr. Christof Strang, Florian Prätsch, Prof. Dr. Thomas Hachenberg); Mannheim, University Medical Centre Mannheim, Department of Anaesthesiology and Surgical Intensive Care Medicine (Dr. Thomas Kirschning, Dr. Thomas Friedrich, Dr. Dennis Mangold); Marburg, University Hospital, Department of Anaesthesiology (Dr. Christian Arndt, Tilo Koch); Mönchengladbach, Kliniken Maria-Hilf GmbH, Department of Cardiology (Dr. Hendrik Haake, Katrin Offermanns); München, Bogenhausen Hospital, Department of Anaesthesiology (Prof. Dr. Patrick Friederich, Dr. Florian Bingold); München, Klinikum Großhadern, Department of Anaesthesiology (Dr. Michael Irlbeck, Prof. Dr. Bernhard Zwissler); München, Klinikum Neuperlach, Städtisches Klinikum München GmbH, Department of Anaesthesiology, Critical Care and Pain Medicine (PD Dr. Ines Kaufmann); München, Klinikum rechts der Isar, Department for Anaesthesiology of the Technical University of Munich (Dr. Ralph Bogdanski, Dr. Barbara Kapfer, Dr. Markus Heim, PD Dr. Günther Edenharter); Münster, University Hospital Münster, Department for Anaesthesiology, Intensive Care Medicine and Pain Therapy, (Prof. Dr. Björn Ellger, Daniela Bause); Neumarkt, Kliniken des Landkreises Neumarkt i.d.OPf, Department for Anaesthesiology and Intensive Care Medicine (Dr. Götz Gerresheim); Nürnberg, General Hospital Nuremberg, Paracelsus Medical University, Department of Emergency Medicine and Intensive Care (Dr. Dorothea Muschner, Prof. Dr. Michael Christ, Arnim Geise); Osnabrück, Marienhospital Osnabrück, Department of Anaesthesiology (PD Dr. Martin Beiderlinden, Dr. Thorsten Heuter); Passau, Klinikum Passau, Department for Anaesthesiology (Dr. Alexander Wipfel); Regensburg, Caritas Krankenhaus St. Josef, Department for Anaesthesiology (Dr. Werner Kargl, Dr. Marion Harth, Dr. Christian Englmeier); Regensburg, Regensburg University Hospital, Department of Anaesthesiology, Operative Intensive Care (Prof. Dr. Thomas Bein, Dr. Sebastian Blecha, Dr. Kathrin Thomann-Hackner, Marius Zeder); Stuttgart, Katharinenhospital, Department of Anaesthesiology (Dr. Markus Stephan); Traunstein, Klinikum Traunstein, Department of Anaesthesiology (Dr. Martin Glaser); Tübingen, Tübingen University Hospital, Eberhard-Karls University Tübingen, Department of Anaesthesiology and Intensive Care Medicine (PD Dr. Helene Häberle); Ulm, Ulm University, Department of Anaesthesiology (Prof. Dr. Hendrik Bracht, Christian Heer, Theresa Mast); Würzburg, University of Würzburg, Department of Anaesthesia and Critical Care (PD Dr. Markus Kredel, PD Dr. Ralf Müllenbach).

Furthermore, we are grateful to previous members of the Regensburg DACAPO study team (medical documentation: Phillip Sebök, study physician: Kathrin Thomann-Hackner), to the members of the Advisory Board of the DACAPO-Study (Prof. Dr. Julika Loss, Prof. Dr. Bernhard Graf, Prof. Dr. Michael Leitzmann, Prof. Dr. Michael Pfeifer, Regensburg, Germany), to our assisting students (Simon Bein, Vreni Brunnthaler, Carina Forster, Stefanie Hertling, Sophie Höhne, Carolin Schimmele, Elisa Valletta), and we are grateful to Monika Schoell for the linguistic revision of the manuscript.

## Author Contributions

**Conceptualization:** Sebastian Blecha, Magdalena Rohr, Frank Dodoo-Schittko, Susanne Brandstetter, Christian Apfelbacher, Thomas Bein.

**Data curation:** Sebastian Blecha, Frank Dodoo-Schittko, Susanne Brandstetter, Christian Karagiannidis, Thomas Bein.

**Formal analysis:** Florian Zeman, Magdalena Rohr, Frank Dodoo-Schittko, Christian Apfelbacher.

**Funding acquisition:** Christian Apfelbacher, Thomas Bein.

**Investigation:** Sebastian Blecha, Magdalena Rohr, Frank Dodoo-Schittko, Susanne Brandstetter, Christian Karagiannidis, Thomas Bein.

**Methodology:** Sebastian Blecha, Florian Zeman, Magdalena Rohr, Frank Dodoo-Schittko, Christian Karagiannidis, Christian Apfelbacher, Thomas Bein.

**Project administration:** Magdalena Rohr, Susanne Brandstetter, Christian Karagiannidis, Christian Apfelbacher, Thomas Bein.

**Software:** Florian Zeman.

**Supervision:** Florian Zeman, Susanne Brandstetter, Christian Karagiannidis, Christian Apfelbacher, Thomas Bein.

**Validation:** Sebastian Blecha, Florian Zeman, Magdalena Rohr, Frank Dodoo-Schittko, Susanne Brandstetter, Christian Karagiannidis, Christian Apfelbacher, Thomas Bein.

**Visualization:** Sebastian Blecha, Florian Zeman.

**Writing – original draft:** Sebastian Blecha.

**Writing – review & editing:** Florian Zeman, Magdalena Rohr, Frank Dodoo-Schittko, Susanne Brandstetter, Christian Karagiannidis, Christian Apfelbacher, Thomas Bein.

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
