## [Decision Letter · Decision Letter 0]

11 Jul 2022

PONE-D-22-09940Association of analgosedation with psychiatric symptoms and health-related quality of life in ARDS survivors: Post hoc analyses of the DACAPO studyPLOS ONE

Dear Dr. Blecha,

Thank you for submitting your manuscript to PLOS ONE. After careful consideration, we feel that it has merit but does not fully meet PLOS ONE’s publication criteria as it currently stands. Therefore, we invite you to submit a revised version of the manuscript that addresses the points raised during the review process.

Several items require clarification and / or revision if the manuscript is to meet criteria for publication in PLOS ONE:

Address all queries posed by Reviewer 1Address all queries posed by Reviewer 2Address all queries posed by Reviewer 3Address all items indicated by tracked changes in the attached Word document (PONE-D-22-09940 [ed])Please submit your revised manuscript by six weeks from the date of this letter.  If you will need more time than this to complete your revisions, please reply to this message or contact the journal office at plosone@plos.org. Please include the following items when submitting your revised manuscript:A rebuttal letter that responds to each point raised by the academic editor and reviewer(s). You should upload this letter as a separate file labeled 'Response to Reviewers'.A marked-up copy of your manuscript that highlights changes made to the original version. You should upload this as a separate file labeled 'Revised Manuscript with Track Changes'.An unmarked version of your revised paper without tracked changes. You should upload this as a separate file labeled 'Manuscript'.

We look forward to receiving your revised manuscript.

Kind regards,

Linda L. Maerz, MD

Academic Editor

PLOS ONE

Journal Requirements:

2. We noticed you have some minor occurrence of overlapping text with the following previous publication, which needs to be addressed:

- https://annalsofintensivecare.springeropen.com/articles/10.1186/s13613-018-0357-y

In your revision ensure you cite all your sources (including your own works), and quote or rephrase any duplicated text outside the methods section. Further consideration is dependent on these concerns being addressed

Reviewers' comments:

Reviewer's Responses to Questions

**Comments to the Author**

1. Is the manuscript technically sound, and do the data support the conclusions?

Reviewer #1: Partly

Reviewer #2: Yes

Reviewer #3: Partly

2. Has the statistical analysis been performed appropriately and rigorously? 

Reviewer #1: No

Reviewer #2: Yes

Reviewer #3: Yes

3. Have the authors made all data underlying the findings in their manuscript fully available?

Reviewer #1: No

Reviewer #2: Yes

Reviewer #3: Yes

4. Is the manuscript presented in an intelligible fashion and written in standard English?

Reviewer #1: Yes

Reviewer #2: Yes

Reviewer #3: Yes

5. Review Comments to the Author

Reviewer #1: The authors presented results from a post-hoc prospective observational study evaluating the association of sedation and analgesia on health-related quality of life (HRQoL) and the risk of psychiatric adverse effects in ARDS survivors 3, 6, and 12 months after discharge from the intensive care unit (ICU). The authors used the patient database from the large prospective cohort study in Germany (DACAPO study). The authors identified that analgosedation did not impact the HRQoL of ARDS survivors. However, in a multivariable analysis, the administration of ketamine during ICU treatment was associated with an increased risk of psychiatric symptoms. The authors are to be commended for evaluating these outcomes and for a well-designed study.

There are a couple of concerns that I believe need to be addressed before publication. Please see the detailed response below.

Perhaps the main concern is the interaction of different sedatives/analgesics used in the management of ARDS patients included in this evaluation. It is common to switch between therapies or administer concomitant sedatives/analgesics to achieve the appropriate level of sedation. It was not clear if these interactions were accounted for. The study results indicate that ketamine was associated with an increased risk of psychiatric symptoms. However, it's not clear if any administration of ketamine was associated with these findings or if a certain cohort of ketamine patients (those who received other sedatives or analgesics) had an increased risk of psychiatric symptoms. For example, patients who received ketamine and midazolam might have had an increased risk of psychiatric symptoms versus patients who received ketamine and propofol. Furthermore, it would be helpful if authors report in detail the breakdown of sedatives and analgesics (number of patients who received one sedative, two sedatives, and three sedatives, etc.)

The other concern in the design of this evaluation is variation in practice. The study utilized the DACAPO database. Different institutions have various sedatives and analgesic dosing regimens, titration protocols goals, and ARDS management guidelines. It would be helpful to account in the multivariable analysis for the type of institutions and variation in practice.

It was not clear from the study's design whether all sedatives and analgesics extracted from the database were accounted for. According to Table 2, none of the patients received commonly used sedatives and analgesics such as dexmedetomidine, fentanyl, morphine, or lorazepam infusion. For clarity, please comment in the study design section whether these variables were included in the DACAPO database or not.

In the results section, report the number of subjects with positive outcomes findings. Specifically, patients with PHQ-9-score ≥5 and PTSS-14-score ≥4.

The multivariable linear regression results focused on ketamine and midazolam. It would be helpful to report the results for other analgosedatives as a supplementary appendix table.

The discussion section of the manuscript might benefit from discussing why the higher prevalence of psychiatric symptoms was detected at 3- and 6-month and not at the 12-month time point.

The conclusion recommends that ketamine be used for a short duration. The findings of this evaluation don't support this recommendation. The study didn't investigate whether the increased incidence of psychiatric symptoms was related to the longer versus shorter duration of use. It might not be appropriate to recommend a shorter duration of ketamine use over a longer duration of use. I would suggest rephrasing the conclusion and recommendations based on the findings of this study.

Minor edits

Page 4: Study design: change DACOPO to DACAPO

Page 4: Sample: change 1.225 patients to 1,225 patients (comma versus period)

Page 5: First paragraph. Clarify the number of patients who died after discharge from the ICU. Figure 1 indicates N=161 versus N=158

Page 5: Consider changing "The kind of analgesia and sedation" to "The type of analgesia and sedation"

Page 5: "For the screening tools, the cut-off values for being at risk of depression were defined as PHQ-9 (≥5) and for PTSD as PTSS-14 [≥45])." The parenthesis in this phrase is confusing. Please update.

Page 7: Please reword "Sedation was often increased with clonidine (73.1%) and midazolam (59.0%)". It's not clear if this refers to an increase in depth/level of sedation or dose of sedatives.

Table 3: Please clarify n=84 in the table legend. Nowhere in the manuscript was a discussion around that number of patients. Please include a detailed explanation of how this number was achieved in the manuscript text.

Reviewer #2: Given that 262 respondents were excluded due to unknown doses and duration of sedation and analgesia application, can the author comment on impact on their conclusions? In other words, would documented dosages of sedatives during their ICU treatment in these respondents changed the conclusion that prolonged ketamine use lowered health quality of life scores?

Reviewer #3: The data was basically exploratory as the investigators used data from the prospective observational nation‑ wide ARDS study across Germany (DACAPO) to investigate the influence of sedation and analgesia on HRQoL and the risk of psychiatric symptoms in ARDS survivors 3, 6 and 12 months after their discharge from the intensive care unit (ICU).

The analysis was routinely performed using univariate and multivariable procedures as noted in the statistical analysis section of the paper. The tables and figures are well formatted and described. The conclusions appear to follow reasonably from the analyses performed . There are some deficiencies noted.

1. Although retrospective and descriptive, the sample size was 134 subjects and some statistical justification should be given for the adequacy of the sample.

2. In Table 3 missing data is noted especially at six months which is about 26% of the sample in some cases assuming a denominator of 84 at most. The authors need to give this some attention and explain how the missing data issue is addressed, if at all. There is no mention in the statistical analysis section of how this is handled.

6. PLOS authors have the option to publish the peer review history of their article (what does this mean?). If published, this will include your full peer review and any attached files.

Reviewer #1: No

Reviewer #2: No

Reviewer #3: No

---

## [Author Response · Author response to Decision Letter 0]

9 Aug 2022

Thank you for the opportunity to revise our manuscript entitled: “Association of analgosedation with psychiatric symptoms and health-related quality of life in ARDS survivors: Post hoc analyses of the DACAPO study”. We are very grateful for the positive review, helpful suggestions, and comments of the three reviewers and you to improve our paper. We have thoroughly modified the manuscript accordingly. Detailed responses to the reviewers’ comments are listed in the Response letter.

---

## [Decision Letter · Decision Letter 1]

6 Sep 2022

PONE-D-22-09940R1Association of analgosedation with psychiatric symptoms and health-related quality of life in ARDS survivors: Post hoc analyses of the DACAPO studyPLOS ONE

Dear Dr. Blecha,

Thank you for submitting your manuscript to PLOS ONE. After careful consideration, we feel that it has merit but does not fully meet PLOS ONE’s publication criteria as it currently stands. Therefore, we invite you to submit a revised version of the manuscript that addresses the points raised during the review process.

We commend the authors for their response to the requests of the reviewers.  Although Revision 1 is substantially improved, additional clarifications and / or revisions are required if the manuscript is to meet criteria for publication in PLOS ONE:

Address all additional queries posed by Reviewer 1Address all additional Journal Requirements requestedAddress all items indicated by tracked changes in the attached Word document (PONE-D-22-09940_R1.docx)

We look forward to receiving your revised manuscript.

Kind regards,

Linda L. Maerz, MD

Academic Editor

PLOS ONE

Journal Requirements:

Reviewers' comments:

Reviewer's Responses to Questions

**Comments to the Author**

1. If the authors have adequately addressed your comments raised in a previous round of review and you feel that this manuscript is now acceptable for publication, you may indicate that here to bypass the “Comments to the Author” section, enter your conflict of interest statement in the “Confidential to Editor” section, and submit your "Accept" recommendation.

Reviewer #1: All comments have been addressed

Reviewer #2: All comments have been addressed

2. Is the manuscript technically sound, and do the data support the conclusions?

Reviewer #1: Yes

Reviewer #2: Yes

3. Has the statistical analysis been performed appropriately and rigorously? 

Reviewer #1: Yes

Reviewer #2: Yes

4. Have the authors made all data underlying the findings in their manuscript fully available?

Reviewer #1: No

Reviewer #2: Yes

5. Is the manuscript presented in an intelligible fashion and written in standard English?

Reviewer #1: Yes

Reviewer #2: Yes

6. Review Comments to the Author

Reviewer #1: The authors updated the manuscript based on the feedback provided. Thank you for allowing me to review the revised manuscript. The authors addressed all the major and minor edit requests. I have two minor feedback points:

• Please update the header with clear labels for easier understanding of the table. Columns 2 and 3 of Table 3 don’t have labels. Also, consider moving “mean (+/-) SD” and “n(%)” to these column labels.

• Consider an alternative word to “refrained” in the conclusion section, “The administration of ketamine to ICU patients with ARDS should be with caution or refrained due to the alternative of other sedative and analgesic drugs”. Consider: The administration of ketamine to ICU patients with ARDS should be with caution or avoided as alternative sedative and analgesic drugs are available.

Reviewer #2: This is an important research about a commonly used sedative in the ICU that would influence patient care and outcomes. Larger prospective study about ketamine would be beneficial to the community for the future.

7. PLOS authors have the option to publish the peer review history of their article (what does this mean?). If published, this will include your full peer review and any attached files.

Reviewer #1: No

Reviewer #2: No

---

## [Author Response · Author response to Decision Letter 1]

13 Sep 2022

see attached file "Response to reviewers"

---

## [Editor Report · Decision Letter 2]

22 Sep 2022

Association of analgosedation with psychiatric symptoms and health-related quality of life in ARDS survivors: Post hoc analyses of the DACAPO study

PONE-D-22-09940R2

Dear Dr. Blecha,

We’re pleased to inform you that your manuscript has been judged scientifically suitable for publication and will be formally accepted for publication once it meets all outstanding technical requirements.

Kind regards,

Linda L. Maerz, MD

Academic Editor

PLOS ONE

Additional Editor Comments (optional):

The authors have addressed the majority of the requested edits and all of the substantive requested edits. 
---

## [Editor Report · Acceptance letter]

14 Oct 2022

PONE-D-22-09940R2 

Association of analgosedation with psychiatric symptoms and health-related quality of life in ARDS survivors: Post hoc analyses of the DACAPO study 

Dear Dr. Blecha:

I'm pleased to inform you that your manuscript has been deemed suitable for publication in PLOS ONE. Congratulations! Your manuscript is now with our production department. 

Kind regards, 

on behalf of

Dr. Linda L. Maerz 

Academic Editor

PLOS ONE